# A Spatio-Temporal Hybrid Neural Network for Crowd Flow Prediction in Key Urban Areas

**Du He [1], Jing Jia [2], Yaoqing Wang [2],\*, Lan You [2], Zhijun Chen [2], Jiawen Li [2], Qiyao Wu [2] and Yongsen Wang [2]**

[1] Hubei Academy of Scientific and Technical Information, Hongshan Road, Wuhan 430071, China; hedu12345@outlook.com

[2] School of Computer Science and Information Engineering, Hubei University, No. 368, Youyi Road, Wuhan 430064, China; jingjia111@outlook.com (J.J.); yoyo@hubu.edu.cn (L.Y.); chenzj@hubu.edu.cn (Z.C.); 202121116013077@stu.hubu.edu.cn (J.L.); 202021116012237@stu.hubu.edu.cn (Q.W.); 202121116013183@stu.hubu.edu.cn (Y.W.)

\* Correspondence: 202121116013006@stu.hubu.edu.cn

**Abstract:** The prediction of crowd flow in key urban areas is an important basis for city informatization development and management. Timely understanding of crowd flow trends can provide cities with data support in epidemic prevention, public security management, and other aspects. In this paper, the model uses the Node2Vec graph embedding algorithm combined with LSTM (NDV-LSTM) to predict crowd flow. The model first analyzes the correspondence between key areas and grid centers, and the Node2Vec graph embedding algorithm was used to extract spatial features. At the same time, considering urban region type, weather, temperature, and other crowd flow data features, the long short-term memory (LSTM) network model was used for unified modeling. The model uses the crowd flow of the previous three days to predict the crowd flow of the next day. The model was evaluated on the 2020 CCF crowd density competition data set. The experimental results show that the NDV-LSTM model can capture the features of the region association digraph and various crowd flow correlation factors well, and the mean square error of the prediction of the crowd flow in key areas is reduced to 1.5194.

**Keywords:** crowd flow prediction; region association digraph; spatio-temporal hybrid neural network; urban crowd flow; deep learning; graph embedding algorithm; crowd flow feature



## 1. Introduction

Urban big data constitute an important information resource for urban economic development. With the continuous development and progress of cities, urban data such as crowd flow, street grid, crowd trajectory, regional information, and weather features are quietly emerging. Urban data collection is an integral activity [1]. By mining and analyzing various types of data, urban computing describes the internal model of urban development and provides decision support for subsequent management and development of the city. The movement track of people produced by activities within the city shows different features in different types of areas, and there are different crowd flow rules [1]. At the same time, urban population distribution and population activities display a regular spatio-temporal dynamic evolution [2]. Population prediction in key urban areas has gradually become a research hotspot for scholars [3–7], which is of great significance and utility for urban epidemic and public security prevention and other emergency needs. We believe that, in the future, applications based on trajectory data will become closely related to people's daily lives [8].

For the problem of predicting crowd flow, the performance of mathematical formulas is not as good as that of neural network models. This is because the size of crowd flow is influenced by various factors, such as weather features and holiday features, which can cause significant fluctuations. The neural network model can capture these features

well. In the early stage, most researchers in the field of crowd flow prediction used statistical learning algorithms or neural network models to solve the problem of crowd flow prediction with complex features [9,10]. Subsequently, many researchers attempted to explore the law of population flow from various aspects to understand the features of crowd activity in different time ranges, such as trend, cycle law, spatial flow law, and other features that can be adopted and modeled by existing models [11]. With the regular division of urban grid regions, a large number of grid-based population flow data are generated correspondingly, providing a new analytical idea for the study of population flow prediction in key urban areas. Some researchers also use the historical dynamic movement trajectory of a population to learn the law of population flow and predict the future movement trajectory of the population [12]. Population flow data often have complex nonlinear correlation features [13]. The generation and change of urban crowd flow data are directly related to human activities [14] and are also affected by complex factors such as weather, geographical location, and environment. However, the common time series model prediction method can only predict the time series of crowd flow, which lacks an exploration of regional spatial correlation, and the single model does not combine well with the spatial attributes of crowd flow. Conventional convolutional neural networks input regular grid partition when extracting spatial features, which does not take into account the flow relationship between grids well.

This paper proposes a spatio-temporal hybrid neural network for population prediction in key urban areas. In this model, the grid where the center of key areas (such as parks, hospitals, stadiums, and other public places) is located represents the crowd flow information of the core of the region [15], and the flow relationship between the grids is converted into the grid connection strength coefficient. The grid connection strength coefficient is used as the weight to construct the region association digraph. Then, the Node2vec graph embedding algorithm is used to extract spatial features. The factors affecting regional crowd flow, such as weather and region type, are mapped as influencing factors to optimize the model. The long short-term memory (LSTM) network model is used to combine spatial and other features to predict the future population flow in urban areas. Experiments on the 2020 CCF population density data set demonstrate the effectiveness of the proposed algorithm.

## 2. Related Work

Recently it was proved that computational techniques, specifically machine learning, have numerous applications in all engineering fields. Azimirad et al. [16] proposed a consecutive hybrid spiking-convolutional (CHSC) neural controller by integrating convolutional neural networks (CNNs) and spiking neural networks (SNNs). Roshani et al. [17] proposed a system to measure both density and velocity of fluids simultaneously. Mozaffari et al. [18] proposed an equal and equitable federated learning (E2FL) to produce fair federated learning models by preserving two main fairness properties, equity and equality, concurrently.

The existing crowd flow prediction methods mainly include: a method based on historical crowd flow statistics, a prediction method based on spatial clustering, a prediction method based on a time series model, and a prediction method based on the neural network combination model.

The method based on historical crowd flow statistics can predict the result of crowd flow in the future by collecting historical data. The autoregressive comprehensive moving average (ARIMA) model [19] is a classic model based on statistical analysis. This model needs to present the time series information as linear correlation features and is not suitable for trend prediction with complex changes, such as the prediction of regional crowd flow. Aiming at the problem that the ARIMA model is affected by the residual and the results are unstable, Shen [20] proposed an improved residual model for short-term crowd flow prediction. Gui [21] proposed a density-based spatial clustering (DBSCAN) method with noise to identify hot spots in different time periods.

In recent years, the neural network-based crowd flow prediction method has made effective progress [22]. Ma et al. [23] mapped the crowd flow data into images for learning through convolutional neural networks and predicted the crowd flow speed within the scope of large-scale networks with high-precision spatio-temporal correlation. Zhang et al. [24] proposed a Deep-ST model, which uses convolutional neural networks (CNN) to rasterize the spatio-temporal data and models the distance dependence, time proximity, cycle, and trend of space. It also adds features such as weather and holidays to predict the urban crowd flow. Xie et al. [25] adopted the multi-scale sequential convolutional network to realize the re-calibration of short dependence and multi-scale temporal pattern features in the temporal data of human crowd flow. On the basis of the previous model, Zhang et al. [26] added a residual module and proposed a spatio-temporal residual network (ST-ResNet), replacing CNN module in the Deep-ST model with ResNet, so that the model can carry out convolutional mining of distant spatial correlation features.

Recently, many scholars have begun to shift their research perspective from multi-level spatial distribution of population [27] to population mobility. Most of the existing studies use national census data, and there is a lack of research on the population flow of urban areas, the relationship between regions, and the discovery of popular areas from the trajectory database [28]. Guo [29] proposed a spatio-temporal cyclic convolutional network (ASTRCNs) model based on the attention mechanism, taking into account various factors affecting regional population flow and conducting unified modeling. Xiong [30] proposed a DCGRu-RF (diffusion convolutional recurrent unit-random forest) model for short-term crowd flow prediction. Xiong firstly used the DCGRU network to learn the spatial correlation features of crowd flow data. Then, the RF model was selected as the predictor, and the nonlinear prediction model was formed and combined with the flow data to learn the timing features. Qiao et al. [31] proposed an adaptive trajectory prediction of moving objects in the case of big data. The model processed position density through massive moving object data, automatically selected parameters according to moving objects, and then output trajectory prediction results. The most existing studies use a single model to predict urban population data, which has achieved good prediction effect. However, due to the poor combination effect with the spatial attributes of crowd flow, there is a lack of comprehensive exploration of the regular features of crowd activities [32]. When extracting spatial features, the traditional convolutional neural network ignores the flow relationship between grids and poorly captures the information features in regional crowd flow data, with some shortcomings in efficiency. In this paper, the region association digraph is built dynamically based on the crowd flow, and the flow relationship between the grids is considered. The spatial features of the graph and the time features of the crowd flow are extracted, and the Node2Vec algorithm is used to integrate various crowd flow data features such as urban area type, area area, weather, and holidays. The embedding of the region association digraph is more expressive, resulting in a higher quality sequence of graph nodes, so as to improve the accuracy of model prediction. Table 1 lists the main symbols and their meanings in this paper.

**Table 1.** The main symbols and their meanings.

| Symbols | Meanings |
|---|---|
| $x_z^d$ | The crowd flow of area d in time period z |
| $f_{1...T}^d$ | The crowd flow sequence of a certain region d in T historical time periods |
| $p$ | The spatial connection of the day |
| $d$ | Date of the day |
| $k$ | The holiday type of the day |
| $e$ | The weather type of the day |
| $l$ | The lowest temperature of the day |
| $t$ | Area type |

**Table 1.** *Cont.*

| Symbols | Meanings |
| --- | --- |
| $m$ | Area of region |
| $f$ | Crowd flow |
| $f_{b1}, f_{b2}$ | Crowd flow of the previous two days |
| $e_b$ | The weather type of the day before yesterday |
| $l_b$ | The lowest temperature of the day before yesterday |
| $k_b$ | The holiday type of the day before yesterday |

## 3. Spatio-Temporal Hybrid Neural Network Model

This paper aims to alleviate the problems identified in the related work above and uses the Node2Vec graph embedding algorithm to learn the spatial attribute features and predict the spatio-temporal features of crowd flow in key areas.

### 3.1. Problem Definition

The crowd flow sequence of key areas is a time series reflecting the change in regional crowd flow. Let $x_z^d$ represent the crowd flow of area $d$ in time period $z$. $f_{1\ldots T}^d = (x_1^d, x_2^d, \ldots, x_T^d)$ represents the crowd flow sequence of a certain key region $d$ in $T$ historical time periods. F $= (f_{1\ldots T}^1, f_{1\ldots T}^2, \ldots, f_{1\ldots T}^D)$ represents the sequence of all crowd flow in $D$ key areas during the period from 1 to $T$.

Given the historical crowd flow F $= (f_{1\ldots T}^1, f_{1\ldots T}^2, \ldots, f_{1\ldots T}^D)$, we want to learn a model that can predict the crowd flow of $D$ areas to be tested during the period $T$ to $T + h$, Y $= (f_{T\ldots T+h}^1, f_{T\ldots T+h}^2, \ldots, f_{T\ldots T+h}^D)$, where $h$ represents the delay between the current time period $T$ and the target time period.

### 3.2. Overall Framework

NDV-LSTM first constructs the region association digraph and then extracts the spatial features using the Node2Vec graph embedding algorithm to study the flow relationship between different regions of the crowd flow. It then optimizes the model by mapping the weather, region type, and other factors affecting the regional crowd flow as influencing factors. The long short-term memory (LSTM) network model is used to predict the future population flow in urban areas by combining the spatial and other features. Figure 1 shows the model structure diagram of NDV-LSTM, whose core is mainly composed of three parts:

- Node2Vec part: Input the spatial region association digraph with time information and extract the high-dimensional feature vector by using the Node2Vec graph embedding algorithm;
- Feature extraction: The feature information of region type, weather, temperature, holiday, and other crowd flow data is mapped into vectors and introduced into the model to obtain the spatial vector matrix;
- LSTM part: Input a spatial high-dimensional feature vector with time information and multi-dimensional features such as region type, weather, temperature, and holidays.

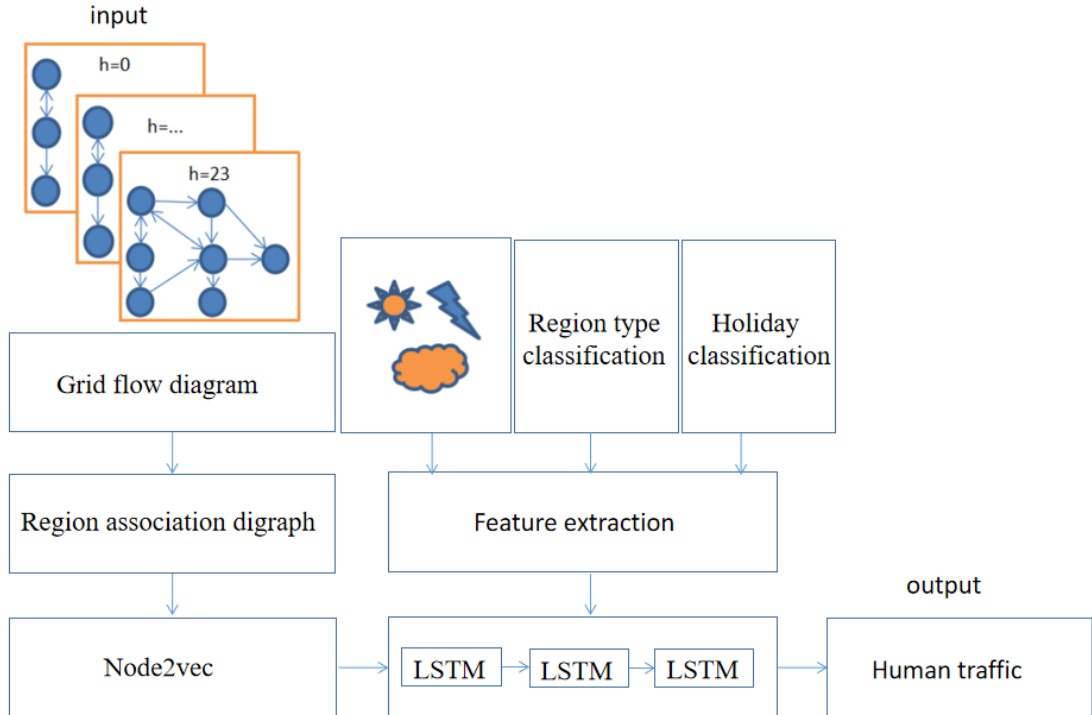

**Figure 1.** NDV-LSTM model frame diagram.

*3.3. Region Association Digraph Design*

In this paper, the grid where the center of key areas (such as parks, hospitals, and other public places) is located represents the core crowd flow of the region, and the data of the regional center grid are used to construct the region association digraph. The crowd flow between the grids is used as the weight of the edges.

Taking F as the input sequence of historical crowd flow, the historical crowd flow F is used to construct a region association digraph, which is formalized as G. As shown in Figure 2, assuming that there is actually a connection between region A and region B at time t, it means that there is certain crowd mobility from place A to B at time t. After constructing the region association digraph, it is converted into the association representation of region A to region B in Figure 3.

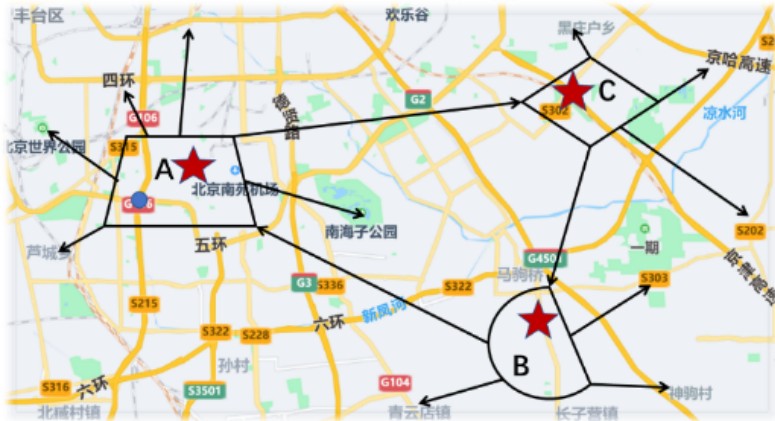

**Figure 2.** Actual region association diagram.

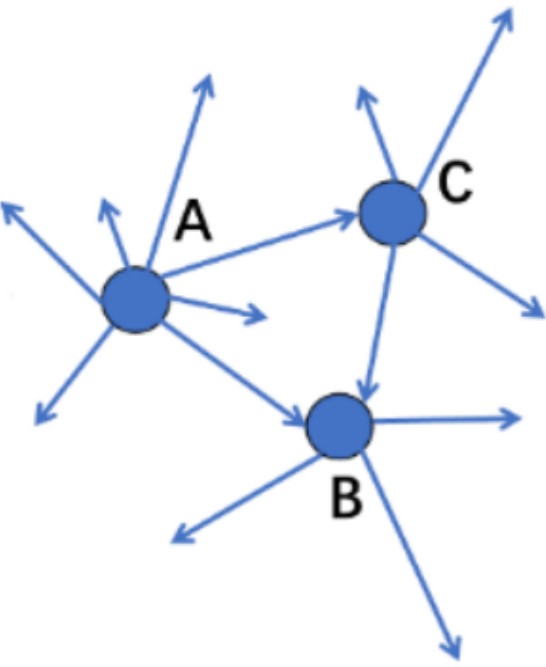

**Figure 3.** Region association diagram.

### 3.4. Region Association Digraph Vectorization

Node2Vec is a random walk graph embedding algorithm that combines depth-first and breadth-first graph sampling [33]. The main idea of the Node2Vec algorithm is as follows: $G$ indicates the association digraph of key areas at a certain time, including the association weight between nodes in the current key area and nodes in adjacent areas. Use the walk sequence generated by all the key region nodes to encode the nodes and obtain the vector representation of each region node. The first key area node is represented by $u$. The Node2Vec algorithm generates a random sequence walk from node $u$, selects the next area node $s$ according to the transfer probability of its surrounding area nodes, adds $s$ to the sequence, and so on. For each node in the region association digraph, this paper sets the probability of the node vector appearing in the adjacent region as the maximum (Equation (1)).

$$\max_{f} = \sum_{u \in V} \log Pr(N_s(u)|f(u)) \tag{1}$$

The probability that each regional node u is selected to join the random sequence is determined by its adjacent node $N(u)$ (Equation (2)).

$$Pr(N_s(u)|f(u)) = \prod_{n_i \in N_{s(u)}} Pr(n_i|f(u)) \tag{2}$$

The probability corresponding to each other region under the node vector of this region is represented by softmax (Equation (3)).

$$Pr(n_i|f(u)) = \frac{exp(f(n_i) \bullet f(u))}{\sum_{u \in V} exp(f(v) \bullet f(u))} \tag{3}$$

After optimization, the objective function is Equations (4) and (5).

$$\max_{f} = \sum_{u \in V} [-log Z_u + \sum_{n_i \in N_{s(u)}} f(n_i \bullet f(u))] \tag{4}$$

$$Z_u = \sum_{u \in V} exp(f(u) \bullet f(v)) \tag{5}$$

Given regional node $u$ and its adjacent regional point $N(u)$, the algorithm maps the vector representation of each key regional node. The final output of Node2Vec is $X_{0......23}$-$Y_{1......997}$=$[p_0, p_1, ......, p_{15}]$, where $X_{0......23}$ is the total vector of the region from 0 to 23 h and $Y_{1......997}$ is the vector under 997 common domains. Therefore, $p$ represents the following Equation (6):

$$p = (p_0, p_1, ......, p_{15}) \tag{6}$$

### 3.5. Description of Features

#### 3.5.1. Holiday Features

There are two forms of urban people's work and rest features: workday and weekend. This paper mainly marks the date as workday and weekend. If day is less than or equal to 5, it is marked as 0 and stored in the workday list, marked as workday. If day is greater than 5, it is marked as 1 and stored in the weekend list, marked as weekend. The form of the holiday feature data is shown in Equation (7):

$$k = (0, 1, 1, ......, 0) \tag{7}$$

#### 3.5.2. Weather Features

In this paper, weather is divided into three categories according to its severity (denoted by 0, 1, and 2 respectively, as shown in Table 1). Suppose $e$ is the weather type and $l$ is the minimum temperature. The form of weather feature data is shown in Equations (8) and (9):

$$e = (0, 1, 2, ......, 2) \tag{8}$$

$$l = (0, -1, -4, ......, -7) \tag{9}$$

#### 3.5.3. Region Information Features

This paper divides 997 public areas into different types according to its region type (parks, zoos, cultural relics, aquariums, scenic spots, botanical gardens, amusement parks, colleges and universities, shopping centers, general hospitals, sports venues, railway stations, airports, long-distance bus stations). $t$ represents the regional types, and $m$ represents the area of the region. The form of the regional information feature is shown in Equations (10) and (11):

$$t = (t_1, t_2, ......, t_{14}) \tag{10}$$

$$m = (m_1, m_2, ......, m_{997}) \tag{11}$$

#### 3.5.4. Feature Fusion

In LSTM, spatial features, holiday features, weather features, and region information features are cyclically combined according to region ID and integrated into multidimensional crowd flow data (Equation (12)) for time series prediction.

$$flowdata = (p, d, k, e, l, t, m, f, f_{b1}, f_{b2}, e_b, l_b, k_b) \tag{12}$$

### 3.6. Prediction of Crowd Flow

As shown in Figure 4, the algorithm adopts two-layer LSTM, and the output of the first layer is used as the input of the second one. The step size of the algorithm is 3, which means that $x_0$ produces the output $x_1$ at the first step, $x_1$ produces the output $x_2$ at the second step, and $x_2$ produces the final output at the third step. Two fully connected layers are used to increase the depth of the model and strengthen the fitting ability of the model. Finally, the dimension of the model is reduced, and the output is the predicted crowd flow.

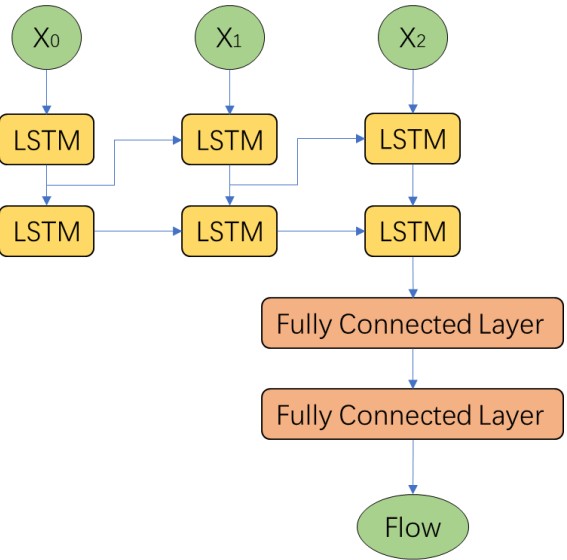

**Figure 4.** LSTM neural network model design diagram.

The input of the LSTM input gate is expressed as Equation (13):

$$w = \sum_{i=1}^{n} s_{ij}x_j^n + \sum_{j}^{c} s_{ij}r_c^{i-1} \tag{13}$$

The input of the output gate is expressed as Equation (14):

$$w_q^n = \sum_{i=1}^{n} s_{ij}x_j^n + \sum_{g=1}^{G} s_{ij}a_g^{i-1} + \sum_{c=1}^{C} s_{cf}a_c^{i-1} \tag{14}$$

The output of the output gate is expressed as Equation (15):

$$a_q^n = \sigma(w_n^q) \tag{15}$$

where $s_{ij}$ represents the weight of neuron $i$ to neuron $j$ in the model, $w$ represents the input, $a$ represents the output, $q$ indicates output gating, $c$ represents neurons, $n$ represents the number of neurons, $g$ represents neuron output activation function, $G$ represents the number of hidden layers of the LSTM, $r$ represents the unit state of the model, and $\sigma$ is the activation function.

The final NDV-LSTM prediction result is expressed as Equation (16):

$$f_{0......997} = (f_0, f_1, f_2, ......, f_{24}) \tag{16}$$

## 4. Experiments

This paper uses the 2020 CCF crowd density competition data set. The data processing and model experiments are carried out in the environment of tensorflow1.14 and python3.7. The algorithm is run in 8G memory, i7-8565U CPU, and Windows 10, including experimental data preprocessing and late feature value processing. This paper mainly uses the software MATLAB to export the charts and diagrams.

### 4.1. Dataset and Pretreatment

4.1.1. Dataset Overview

The crowd flow index in the data set represents the volume of crowd flow in the key area during a certain hour of the day. For example, the larger the crowd flow index of an area R, the more people appear in the area R, and vice versa. The dataset overview is shown in Table 2.

**Table 2.** Dataset overview.

| Field Name | Example |
| --- | --- |
| Region ID | 1 |
| Data-hour | 2020-02-05-10 |
| Crowd flow index | 1.8 |
| Area name | Beiwu Park |
| Area type | Tourist attractions: park |
| Longitude of regional center point | 116.256586 |
| Latitude of regional center point | 39.986913 |
| Grid longitude of regional center | 116.256713 |
| Grid latitude of regional center | 39.987525 |
| Area of region | 395,925.609375 |
| Hour | 0 |
| Longitude of departure grid center | 115.99432 |
| Latitude of departure grid center | 39.698475 |
| Longitude of destination grid center | 115.99432 |
| Latitude of destination grid center | 39.698475 |

4.1.2. Experimental Data Processing

(1) Spatial feature data preprocessing: As can be seen from the previous section, the 24 files in the area-flow-graph folder all correspond to the region-to-region association in different hours. The region association digraph is obtained, and then the spatial features of the region information are extracted to obtain the vector representation.

(2) Sequential feature data preprocessing: The algorithm sets a 24-h cycle, and there are 30 days' worth of historical data in each cycle. The experimental date data is from the 17th to the 15th of the next month, and the 17th is marked as 0, while the 15th of the next month is marked as 29. Read the file area: Passenger—Regional crowd flow information for 30 days. The area-flow list stores (ID, day-hour, crowd flow index) information.

(3) External influence feature data preprocessing: This includes region information, weather features, and holiday feature data preprocessing. The regional category is divided into tourist attractions, education training, shopping, medical treatment, sports fitness, and transportation facilities. Weather features are divided into moderate snow, cloudy, light rain, light snow, fine, blizzard, overcast, sleet, fog, and haze. Holiday feature data are mainly marked as workdays and weekends.

A variety of data features are fused into a fixed format and stored in the flow-data list, and then the regional crowd flow is normalized to the range [0, 1] by using the min-max method.

The model divides the data into 24 h blocks for cycle training. During each hour of training, the flow-data list contains the crowd flow data of 997 areas in the past 30 days during this period. There are 717,840 pieces of data in this experiment, with the training set accounting for 90% and verification set for 10%. The data from a certain hour in the previous three days are used to predict the result of an hour in the next day, and the final circulation output is 24 h in the next day.

*4.2. Baseline and Evaluation Index*

In this paper, the region association digraph is constructed to learn spatial features. In order to verify the effectiveness of the NDV-LSTM model in predicting crowd flow in key areas, existing model methods HA, ARIMA, and LSTM are selected for comparative experiments and ablation experiments. The comparison model is as follows:

(1) Historical average (HA): The idea of this model is using the 24-h crowd flow index of the previous three days to predict the 24-h crowd flow index of the next day. Therefore, the HA method takes the average of a certain hour in the previous three days as the crowd flow of the area in the next hour.

(2)　Autoregressive integrated moving average (ARIMA) model: This is a classical method to predict time series in various regions. The paper adopts the round-robin prediction method to obtain the prediction results under 24 h and average them as the final ARIMA error results.

(3)　Long short-term memory (LSTM) network model: The model itself has great advantages in time series prediction.

(4)　NDV-LSTM-type: The model removes the spatial region type feature.

(5)　NDV-LSTM-day: The model removes the rest feature.

(6)　NDV-LSTM-weather: The model removes the weather type feature.

In order to assess the prediction effect of the NDV-LSTM model, the mean square error (MSE) of the model prediction results under 24 h was selected as the evaluation and comparison index of the model experiment. The MSE formula is as follows (Equation (17)):

$$MSE = \frac{1}{n} \sum_{i=1}^{n} (y_i - \overline{y_i})^2 \tag{17}$$

where $yi$ represents the predicted result value of the model and $\overline{y_i}$ represents the real label value of the source data [34].

*4.3. Experimental Result*

The final experimental results are shown in Table 3.

**Table 3.** Baseline experimental results.

| Model | MSE |
|---|---|
| HA | 4.4879 |
| ARIMA | 13.4669 |
| LSTM | 9.2632 |
| NDV-LSTM-type | 3.1141 |
| NDV-LSTM-day | 2.6062 |
| NDV-LSTM-weather | 3.6113 |
| NDV-LSTM | 1.5194 |

*4.4. Experimental Analysis*

4.4.1. Model Comparison Analysis

Through the above experimental comparison results, it can be seen that:

(1)　The mean square error of the HA method is relatively small. The algorithm uses the crowd flow of each hour on February 12, 13, and 14 to predict the crowd flow of this region on February 15. Among them, February 13, 14, and 15 fall on Tuesday, Wednesday, and Thursday, so the results of adjacent workdays are relatively stable.

(2)　The mean square error of the ARIMA method is relatively large because it needs to input a stable time series and cannot learn the features of complex factors. There are holidays from January 17 to February 25, and the sequential features of crowd flow in the early stage are not stable, resulting in poor performance.

(3)　The mean square error of the LSTM neural network model is large due to spatiotemporal data that are related to a variety of factors; there are many factors involved in the features of crowd flow. The time series data of the experiment is from the Spring Festival to workdays, and the weather in Beijing is cold. Therefore, it is difficult to capture normal and stable time sequence rules to predict the future crowd flow in key areas. The overall training results of the NDV-LSTM model proposed in this paper are stable, the prediction results are more accurate with the input of different features.

In this paper, the ablation experiment was conducted, and the influence of different types of feature data on the experimental results was analyzed. The mean square error decreases by 51% after removing the region type feature (NDV-LSTM-type), which proves

the necessity of the actual region type feature for the prediction of the crowd flow in key areas. After removing the holiday feature (NDV-LSTM-day), the mean square error decreases by 57%, which proves that the work and rest type features have an impact on the prediction of crowd flow. After removing the weather type feature (NDV-LSTM-weather), the mean square error decreases significantly because the data set was collected in winter; therefore, the weather type feature has a great influence on the model prediction results.

### 4.4.2. Parameter Settings

The mean square error results of the NDV-LSTM obviously decrease with high training epochs and drop to the lowest when the epochs total 1000. As shown in Table 4, the epochs are set at 1000, and the MSE is 1.5677.

**Table 4.** Comparison of the results under different epochs.

| Model | Epochs | MSE |
| --- | --- | --- |
| NDV-LSTM | 100 | 54.4532 |
| | 500 | 14.6607 |
| | 800 | 2.7467 |
| | 1000 | 1.5677 |

Under the same time period and epoch but different batch sizes, the model training error is not much different, as shown in Table 5. The MSE is decreased to 2.6433 when the batch size is 128.

**Table 5.** Comparison of the results under different batch-sizes.

| Model | Batch-Size | MSE |
| --- | --- | --- |
| NDV-LSTM | 32 | 2.6450 |
| | 64 | 2.6433 |
| | 128 | 2.6433 |

### 4.4.3. Analysis of Crowd Flow in Key Areas

In order to more intuitively analyze the rationality of modeling and understand the actual law of crowd flow in key areas, this paper analyzes the prediction results of crowd flow in 3 adjacent days during the same hour in the same area type. For example, we can choose three tourist attractions—Beiwu Park, Guta Park, and Niantan Park—and obtain the crowd flow at 2:00 p.m. on February 13, 14, and 15. Figure 5 represents the real data, and Figure 6 represents the predicted result data of NDV-LSTM. The horizontal axis is the date, and the vertical axis is the region ID. The scale on the right means that the darker the color, the greater the amount of crowd flow in the thermodynamic diagram. It can be seen from the figure:

(1) If the region type is the same, the crowd flow index is similar in the same hour on three adjacent days. This also verifies the validity of the modeling idea of using the same hour of the previous three days to predict the crowd flow of the next day.

(2) It can be seen that the predicted results of the model are also consistent with the actual situation of the flow trend of the first three days in the same hour and the same area type, thus verifying the accuracy of the predicted results of the model.

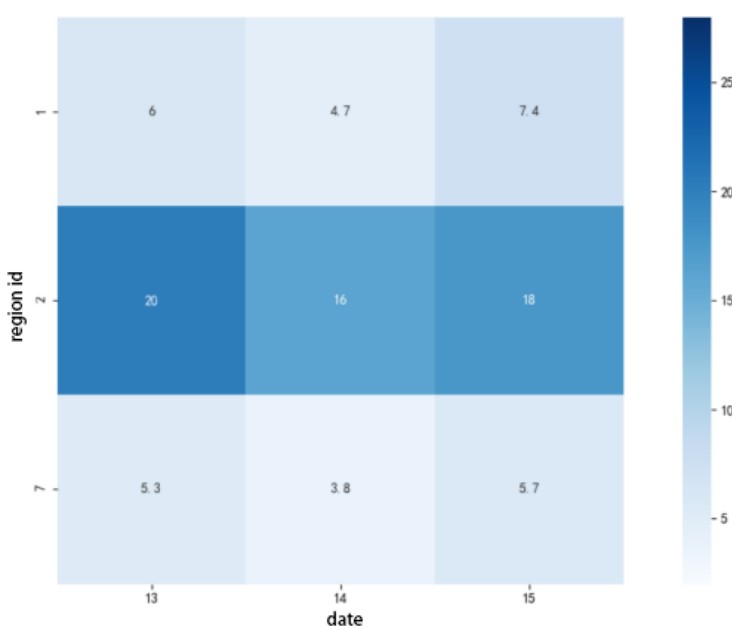

**Figure 5.** Thermodynamic diagram of actual crowd flow.

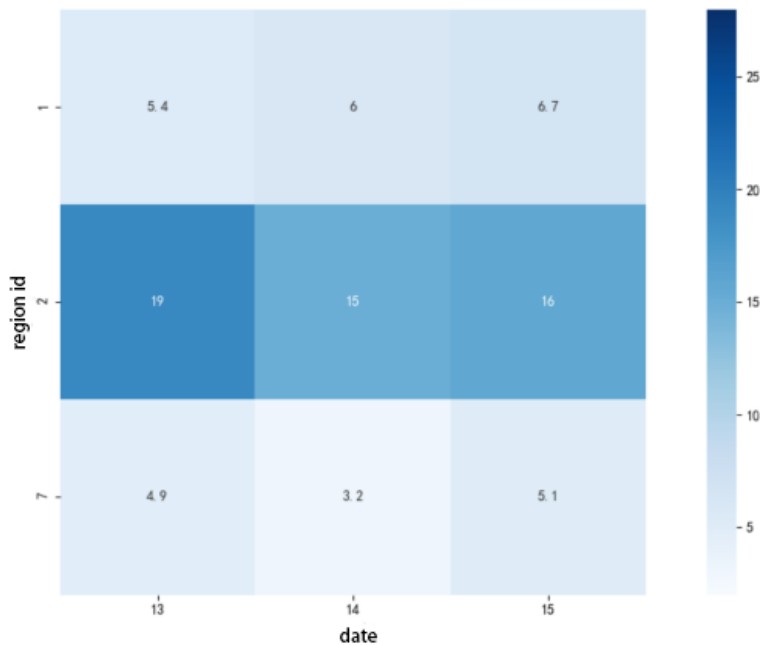

**Figure 6.** Thermodynamic diagram of predicted crowd flow.

## 5. Conclusions and Future Work

In view of the high spatio-temporal correlation of the crowd flow, and combining various factors affecting the prediction of urban crowd flow, this paper proposed a prediction model of urban crowd flow based on a multi-dimensional hybrid neural network model. Different from other models in regular European grid space, this paper constructed a region association digraph based on grid connection strength. The Node2Vec graph embedding algorithm was used to transform the region association digraph under different key regions into a multi-dimensional vector sequence. At the same time, multiple factors were mapped into the LSTM model to study and predict the crowd flow in key areas. The experimental results showed that the introduction of region type, holiday, weather, and other features can significantly reduce the error of prediction, which proves the validity of the NDV-LSTM model prediction.

In this paper, data analysis and reasonable application were carried out for the prediction of crowd flow in key urban areas. The accuracy of the model was improved to a certain extent, but there is still room for improvement as follows:

(1) The normal flow of people has complex migration and other rules. How to consider the multiple features of the law of crowd flow more comprehensively will continue to be studied.

(2) The crowd flow relationship and the features of the crowd flow in adjacent hours will also have an impact on the prediction. How to let the model learn the feature information of adjacent hours is also the focus of subsequent research. We will try using an attention mechanism with LSTM and encoder-decoder architecture to improve the performance for the future improvement.

**Author Contributions:** Conceptualization, methodology, funding acquisition, D.H.; Validation, formal analysis, J.J.; Writing—review and editing, experiment, Y.W.(Yaoqing Wang); Investigation, L.Y.; Original draft, Z.C.; Experimental analysis, conclusion, J.L.; Data collection, Q.W.; Resources, data curation, Y.W.(Yongsen Wang). All authors have read and agreed to the published version of the manuscript.

**Funding:** This work was partially supported by the Technology Innovation Special Program of Hubei Province (No. 2022BAA044, No. 2021BAA188), the Key Project of Science and Technology Research Program of Hubei Provincial Education Department (No. D20201006), and the National Natural Science Foundation of China (No. 61977021).

**Institutional Review Board Statement:** This study did not involve humans or animals.

**Informed Consent Statement:** This study did not involve humans.

**Data Availability Statement:** All the details of this work, including data and algorithm codes, are available from the corresponding author: 202121116013006@stu.hubu.edu.cn.

**Acknowledgments:** The authors would like to thank the reviewers for their helpful suggestions, which have considerably improved the quality of the manuscript.

**Conflicts of Interest:** The authors declare no conflict of interest.

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
