# Peer review of "A Spatio-Temporal Hybrid Neural Network for Crowd Flow Prediction in Key Urban Areas"

_electronics, doi:10.3390/electronics12102255_

Round 1

Reviewer 1 Report

The manuscript entitled “A Spatio-temporal Hybrid Neural Network for Crowd Flow Prediction in Key Urban Areas” has been investigated in detail.
The paper’s subject could be interesting for readers of journal. Therefore, I recommend this paper for publication in this journal but before that, I have a few comments on the text that should be addressed before publication:

·         Some abbreviations are missing in the text, please check the manuscript thoroughly to find the ones and add them in the table.

·         Please mention clearly in the text that what is the main benefit of using artificial intelligence over mathematical equations for such problems.

·         How did you evaluate the accuracy of used neural networks?

·         Which software has been used in this work to export the charts and diagrams in this work? For instance, software like SigmaPlot or SmartDraw are used to export and depict charts. Mentioning used software would be helpful to future researches and studies in the field of this article.

·         The problem statement is not explained comprehensively. What is the novelty of present investigation?

·         Mentioned errors in the conclusion section must be compared with the previous studies in the field

·         Since recently it has been proved that computational techniques, specifically machine learning has a numerous applications in all of engineering fields, I highly recommend the authors to add some references in this manuscript in this regard. It would be useful for the readers of journal to get familiar with the application of computational techniques in other engineering fields. I recommend the others to add all the following references, which are the newest references in this field

[1] Azimirad, V., Ramezanlou, M. T., Sotubadi, S. V., & Janabi-Sharifi, F. (2022). A consecutive hybrid spiking-convolutional

(CHSC) neural controller for sequential decision making in robots. Neurocomputing, 490, 319-336.

Suspensions. Physical Review Letters, 129(6), 068001."

[2] Roshani, et al., 2018. Density and velocity determination for single-phase flow based on radiotracer technique and neural networks. Flow Measurement and Instrumentation, 61, pp.9-14

[3] Mozaffari, H., & Houmansadr, A. (2022). E2FL: Equal and Equitable Federated Learning. arXiv preprint arXiv:2205.10454.

Author Response

Thank you for your review and comments of our manuscript. We revised the paper according to the two reviewers’ comments. The file is the point to point response to your comments.

Best Regards,

Authors

Reviewer 2 Report

The study presents very interesting topic for urban area and traffic with useful results. Node2vec with LSTM were implemented for prediction urban crowd flow in key urban areas with feature information like regional type, weather, temperature, holiday rest etc.

I didn't notice any serious mistakes, but I have a couple of suggestions for the authors that are non-binding:

- It would be desirable to put in the abstract how long the model predicts (time step).

- I did not notice in the text that the abbreviation NDV was defined.

- Page 3 lines 125 and 131: At first glance I thought that it was a minus f1-T and fT-(T+h) but it is range, from 1 to T so maybe put something else and not minus?

- Page 6 line 201: t represents regional types but on page 3 line 125 t is defined as time period t. Maybe change?

- Page 7 line 212: The step size of the algorithm is 3,

It would be useful to define The step size.

- Page 9: The model divides the data into 24 hours for cycle training. During each hour of training, the flow-data list contains the crowd flow data of 997 areas in the past 30 days during this period. There are 717,840 pieces of data in this experiment.

Does this mean that you trained the model for one month and not for 12 months? Do you think that you capture seasonality with external features that you implemented?

- Figures 5 and 6: It would be good to define the scale on the right, what does it represent?

- Can you explain how you put features from Table 1 in LSTM in few sentences?

- Page 7 Table 1 and equations (13) - (15): variables w (greek small omega), s and are defined differently. Change?

- Page 12 line 353: for future improvement (2) my advice is to try attention mechanism with LSTM and encoder-decoder architecture.

Minor editing of English language required

Author Response

(The authors gave the same response as above.)

Reviewer 3 Report

Very good article!! It concerns the possibility of city management in the smart city system and the possibility of predicting crowd flow trends in key urban areas. Timely understanding of crowd flow trends can provide cities and urban areas with data support in city life, security management, public safety management, or traffic relief, spatial factor control. The article proposes a spatio-temporal hybrid neural network for population prediction in selected urban areas. The strength of the article is a well-developed methodology and an excellent idea for crowd flow modeling and modeling it. Very good conclusion!! In principle, the article could be printed as presented.

Comments:

1. list of abbreviations - please do at the beginning of the article

2. abstarct - definitely too long, please shorten it, because it discourages reading. The abstract should contain only the most important assumptions and the most important research results

The article fits very well in the Electronics journal profile

Author Response

(The authors gave the same response as above.)

Round 2

Reviewer 1 Report

Comments have been addressed carefully. I believe the paper is ready for acceptance in the present form.